# Using Compound Neural Action Potentials for Functional Validation of a High-Density Intraneural Interface: A Preliminary Study

**DOI:** 10.3390/mi15020280

**Published:** 2024-02-17

**Authors:** Aritra Kundu, Erin Patrick, Seth Currlin, Ryan Madler, Francisco Delgado, Ahmed Fahmy, Rik Verplancke, Marco Ballini, Dries Braeken, Maaike Op de Beeck, Nima Maghari, Kevin J. Otto, Rizwan Bashirullah

**Affiliations:** 1Department of Bioengineering, Imperial College London, SW7 2AZ London, UK; 2Department of Electrical and Computer Engineering, University of Florida, Gainesville, FL 32611, USA; erin.patrick@ece.ufl.edu (E.P.); maghari@ufl.edu (N.M.);; 3Department of Biomedical Engineering, University of Florida, Gainesville, FL 32611, USAkevin.otto@bme.ufl.edu (K.J.O.); 4Centre for Microsystems Technology (CMST), IMEC and Ghent University, 9052 Zwijnaarde, Belgiummaaike.opdebeeck@imec.be (M.O.d.B.); 5TDK InvenSense, 20057 Milan, Italy; 6IMEC, Kapeldreef 75, 3001 Leuven, Belgium; dries.braeken@imec.be; 7Galvani Bioelectronics, South San Francisco, CA 94080, USA

**Keywords:** intrafascicular electrodes, compound nerve action potentials, peripheral nerve, rat model

## Abstract

Compound nerve action potentials (CNAPs) were used as a metric to assess the stimulation performance of a novel high-density, transverse, intrafascicular electrode in rat models. We show characteristic CNAPs recorded from distally implanted cuff electrodes. Evaluation of the CNAPs as a function of stimulus current and calculation of recruitment plots were used to obtain a qualitative approximation of the neural interface’s placement and orientation inside the nerve. This method avoids elaborate surgeries required for the implantation of EMG electrodes and thus minimizes surgical complications and may accelerate the healing process of the implanted subject.

## 1. Introduction

### 1.1. Peripheral Nerve Interfaces

Peripheral nerve interfaces, used in conjunction with prosthetic limbs, are designed to record from efferent and/or stimulate afferent neural fibers to provide communication to re-establish the lost functionality of missing limbs [1,2]. In the design of peripheral nerve interfaces, a trade-off between increasing the invasiveness to obtain adequate selectivity among multiple nerve fascicles/fibers and minimizing the invasiveness to provide ease of implantation and reduce the foreign body response, which may have an impact on the chronic stability of the interface [3], exists. In this light, two categories of nerve interfaces are common in the literature: extraneural and intraneural electrodes [3,4,5].

Over the years, extraneural electrodes, including cuff, epineural, and FINE [6,7,8,9,10,11,12,13,14,15,16,17,18], and intraneural electrodes, e.g., LIFE, tfLIFE, TIME, and USEA [19,20,21,22,23,24,25,26,27,28,29,30,31,32,33,34,35,36,37,38,39], have been designed and tested in both animals and humans in different conditions with various degrees of success [3,5,40]. Although extraneural electrodes are less invasive and simple to handle, they have limited selectivity, i.e., the electrode contacts most often communicate with the fascicles lying close to the surface of the nerve [32]. Intraneural electrodes implanted inside the nerve, though invasive, have a high signal-to-noise ratio and a larger range of stimulation and recording selectivity [41]. The human nerve topography shows the innervation of different skeletal muscles by the fascicles lying both near the periphery and at the central regions of the nerve [42,43]. Therefore, if implanted precisely, an intraneural electrode will have the capability to interface with fascicles in different regions. Moreover, intraneural electrodes have the additional advantage of a lower stimulation current and an increased signal-to-noise ratio in their recordings compared to extraneural electrodes [20,30,32]. The common intraneural electrodes that can be placed intrafascicularly to date are LIFE, tfLIFE, TIME, and USEA [2,4].

### 1.2. High-Density Transverse Intrafascicular Multichannel Electrode and Integrated Electronics

To maximize the performance of an intraneural interface, a novel high-density transverse intrafascicular multichannel electrode (hd-TIME) has been designed and fabricated [44]. The hd-TIME probe consists of a CMOS electronic chip thinned and embedded into a flexible biocompatible polyimide substrate for transverse insertion into human nerve fascicles for both recording and stimulation. The CMOS chip includes recording and stimulation sites and amplification and multiplexing circuitry for added functionality. The highest density implanted probe contains 64 neural recording electrodes and 16 fully reconfigurable neural stimulating electrodes. For more information about the fabrication, type, and functionality of the probe, read [44,45]. The stimulation modality of the hd-TIME will be used to encode sensory percepts via the evoked stimulation of the afferent nerve fibers within a fascicle. 

“Non-remote” integrated electronics [46] help to reduce the wiring and signal loss of transmission. Miniature active electronics, along with lead electrodes [47], reduce the footprint and will be a minimum challenge to implant into human-sized nerves. Therefore, it has been envisioned that hd-TIME will have an advantage over low-channel-count intraneural electrodes.

### 1.3. Metrics for Functional Assessment

The stimulation efficacy of earlier generations of multichannel intrafascicular electrodes like tfLIFE and TIME has been investigated using the measured response of muscle activity, with electromyograms (EMGs) as the metric for quantification [29,32,34,35,41,48]. The quantification method consists of measuring the evoked EMG response in specific muscles for a range of stimulus parameters (e.g., cathode first, a biphasic current pulse with a constant pulse width, and increasing pulse amplitudes). With this method, the power of the EMG response (calculated using the peak-to-peak or RMS values of the EMG signal) is plotted with respect to the stimulus’ current amplitude, for example, resulting in a sigmoidal curve that relates a minimal and maximal response to the electrical stimulation. Researchers use this curve as an estimate of the neural fiber recruitment, as the EMG response correlates with the number of fibers activating the muscle [49]. Although neural fiber recruitment estimation using EMGs is widely used, it does not contain information about the sensory fiber activation. Since our interface is intended to be used to provide sensory perception via electrical stimulation of the afferent nerve fibers, a quantification metric that directly measures the response of the afferent populations of interest is needed. 

Compound nerve action potentials (CNAPs) are the summation of individual axonally propagating action potentials arising from a stimulus. Electrical stimulation of the nerve can evoke action potential propagation in the efferent and afferent fibers; thus, the CNAPs arising from electrical stimulation contain information on the excitability of all the fibers in the nerve. 

To the best of our knowledge, limited studies have employed only CNAP-derived recruitment curves to assess the stimulation efficacy over a wide range of stimulus parameters for intraneural peripheral nerve interfaces. Because of this fact, we evaluate an experimental method using measured CNAPs that can be used for acute and chronic in vivo assessment of the stimulation effectiveness of the hd-TIME. 

Moreover, in surgeries performed on large animal models or even humans, measuring the CNAPs will mitigate the additional surgeries needed to implant EMG electrodes into the muscles of a limb. The strength of CNAPs is generally independent of the distance between the stimulating and recording electrode, whereas the strength of EMGs depends on the position on the muscle. Thus, they can give a false indication of the stimulation strength. Crosstalk, i.e., contamination from muscles lying deeper under the muscle of interest while recording, is also avoided [50]. The objective of this work is to evaluate an experimental methodology to measure the stimulation selectivity using only measured CNAPs for acute in vivo assessment of a passive version of the hd-TIME. 

## 2. Materials and Methods

### 2.1. Animal Preparation

All the experimental procedures were approved by the University of Florida Institutional Animal Care and Use Committee (IACUC) guidelines. Anesthesia was induced and maintained for the duration of surgery with 1–3% isoflurane in oxygen at 1–2 L/min. Upon induction, all rats (Lewis rats, Charles River Laboratories, Wilmington, MA, USA) received preoperative meloxicam (1–2 mg/kg SQ, Loxicom, Norbrook Laboratories, Newry, Northern Ireland). The surgical sites were shaved using electrical clippers and aseptically prepped for surgery. The rats were prone-positioned on a circulating water bath/heating pad to maintain their core body temperature under a surgical microscope (V8 Stereomicroscope, Zeiss; Jena, Germany). Their heart rate and hemoglobin oxygen saturation were monitored continuously throughout the procedure (PhysioSuite, Kent Scientific, Torrington, CT, USA). Eye ointment was applied to both eyes to prevent them from drying and becoming irritated.

The sciatic nerve was assessed by making a 3–4 cm long cutaneous incision. It was made over the lateral aspect of the left limb or right limb or both, over the femur. The sciatic nerve was exposed by bluntly dissecting the muscle bellies of the biceps femoris and gluteus maximus. The nerve was exposed proximally to the iliofemoral ligament and distally (1 cm) after the bifurcation of the sciatic nerve into the tibial, peroneal, and sural branches. They were carefully freed from the connecting tissues. 

Three rats (Rat1, Rat2, and Rat3) were implanted with the hd-TIME, into the sciatic nerve in either the left or right limb or both, making a total of *n* = 5 stimulation studies. 

### 2.2. The Electrodes Implanted 

The design and fabrication of an hd-TIME were explained in detail in [44]. They are laborious and expensive processes, raising the manufacturing cost of each probe. Each step of fabrication needs to be meticulously executed. De-risking the hd-TIME becomes essential to avoid design and functional glitches in the definitive version. A passive version of the hd-TIME has been fabricated to optimize the design and implantation strategy. The passive probe of both variants has only four working stimulation sites and is devoid of any active circuitry. In the following sections, reference to the hd-TIME refers to the passive short version of the probe. Stimulation sites 0, 5, 10, and 11 are the functional electrodes of the probe (Figure 1). There are 10 working recording sites, but the scope of this current study is limited to stimulation. A ground or reference electrode is also present in the probe many millimeters away from the stimulation/recording electrodes (see Figure 1).

The hd-TIME was implanted transversally into the proximal sciatic nerve of the rats, approximately 5 mm from the bifurcation (see Figure 2). Stimulation pulse trains of different amplitudes were injected, and CNAPs were recorded from the three branches, i.e., tibial (T), peroneal (P), and sural (S), using cuff electrodes. Since the hd-TIME is an intrafascicular electrode, ideally, each stimulation site should be able to selectively activate a sub-population of nerve fibers, a fascicle, or a small sub-group of fascicles in a poly-fascicular nerve. The inbuilt ground electrode was used for the ground stimulation, making sure that it made contact with the tissue inside the surgical opening but was not inside the nerve. 

The recording tibial and peroneal cuffs are of a 750 µm and 500 µm diameter, respectively, and were fabricated at Microprobes (Microprobes for Life Science, Gaithersburg, MD, USA). Meanwhile, a 500 µm cuff was used for the sural nerve and was fabricated at CorTec (CorTec GmbH, Freiburg, Germany). A separate external ground electrode was placed on a nearby muscle belly.

During these studies, the minimum distance between the stimulating hd-TIME and each recording cuff electrode was between 10 mm and 16 mm in every study due to the restrictions of the length of the surgical opening in a rat model. 

### 2.3. The Stimulation Paradigm

Biphasic, cathodic-first, 25 µs inter-pulse duration stimulation pulses were used for charge balanced stimulation, and their amplitude comprised a range of 2 to 650 µA. The pulse train amplitudes were delivered in pseudo-random order to reduce the effect of neural adaptation. The pulse train was repeated multiple times (35 to 40) and was delivered at 5 Hz to the electrode using an isolated IZ2H Stimulator (Tucker-Davis Technologies, Alachua, FL, USA) (Figure 2). The stimulation was repeated for 50 µs and 100 µs pulsewidths. 

### 2.4. Data Collection and Analysis

The CNAPs obtained from the cuff electrodes from the three branches were recorded at a 48 kHz sampling frequency using a 32-channel PZ5 NeuroDigitizer optically coupled to an RZ5D recording bio-amplifier (Tucker-Davis Technology, Alachua, FL, USA) (Figure 2). Time segments referenced to the stimulation time stamps were used to average the CNAP waveforms and calculate the recruitment curves. The data analysis was performed offline using MATLAB (MathWorks, Natick, MA, USA). To quantify the level of CNAP recruitment, the area under the curve was calculated in a variable time window for each nerve.

### 2.5. Analysis of the Selective Recruitment of Nerve Fascicles

The area under the curve of the CNAPs was calculated to quantify the stimulation. The TRAPZ function in MATLAB was used. The values were normalized by the largest value across both pulse widths for each nerve. 

The range in the calculation of CNAPs consisting of both peaks and troughs for any nerve lies between 594 µs and 1044 µs. The area under the curve (AUC) was chosen over the peak-to-peak voltage and RMS. Due to multiple repetitions of stimuli for each amplitude, the noise from the electrophysiological recording is minimized by averaging the stimulus–response. Any DC offset was corrected by averaging the first 12–14 data values before the stimulus–response and then subtracting the entire stimuli and its response from the averaged value. This is necessary to avoid overprediction or underprediction of the CNAPs’ area under the curve values. Digital filters were avoided, as they introduce unwanted ripples into the signal that might be mistaken for CNAPs, or contamination of the CNAPs.

CNAP data were obtained from all three sciatic nerve branches. The area under the curve was measured, and the recruitments were plotted. Their recruitment values were normalized to the highest value across all three nerve branches (T, P, and S) and over both 50 µsec and 100 µsec pulse widths. The nature of recruitment and the number of nerve branches that were recruited were visualized. No score or formula was used. Recruitment of the tibial branch fibers will result in some recruitment of the sural branch fibers, as the sural nerve branches from the tibial [51]. Therefore, it is highly probable that activating the tibial will automatically activate the sural. 

### 2.6. Predicting the Electrode Location Inside the Nerve

Based on the nerve branch recruitment, visual inspection, the electrode stimulation site, and the direction of the implant (left or right), the location of the stimulation site was estimated, and schematic drawings were made for each hd-TIME implant (Figure 10). 

## 3. Results and Discussion

### 3.1. CNAPs with Respect to Stimulation Artifacts and Parasitic EMG Signals

One of the challenges when recording CNAPs is the contamination of the signal by the stimulus artifact, as well as electromyogram (EMG) signals from activated muscles. Figure 3 shows an exemplary recording where the CNAP, which is the smallest amplitude feature, is in between the stimulus artifact and the EMG signal. This CNAP signal was recorded using a monopolar cuff electrode with a distant ground on the tibial branch of the left sciatic nerve in one of the animals. This recording scheme results in large-amplitude CNAPs and thus a good signal-to-noise ratio compared to bipolar or tripolar recording schemes, but it also does not subtract out any of the unwanted signals, such as the stimulus artifact or EMG. The dashed lines in Figure 3 correspond to the recorded signals for various stimulus amplitudes after the nerve was transected distal to the recording cuff electrode. The CNAP signal changed only slightly after the removal of the EMGs via nerve transection, showing that the CNAP is indeed the second feature after the stimulus artifact and can be influenced by the presence of EMGs. 

The distance between the stimulation and recording sites is of immense importance for the detection of single or compound nerve action potentials [52]. When the nerve is not dissected from muscle, if the distance between the stimulation and recording electrodes is too long, then chances are the EMGs will engulf the CNAPs, whereas shorter distances between two electrodes result in part or whole of the CNAP being lost within the stimulation artifact itself (Figure 4). If an ideal distance is maintained, the CNAPs occur a few microseconds after the stimulation artifact. Maintaining this ideal stimulation–recording electrode distance can be a bit challenging in small animal models due to a lack of space. In the case of Figure 3, the stimulation-to-recording distance was near 16 mm. Figure 5 shows the experimental results in a case where the stimulation-to-recording distance was similar to the one in Figure 3 (e.g., about 16 mm), except the pulse width was much larger (500 μs compared to 50 μs) and the nerve was not transected distal to the cuff recording site. In this case, the CNAP recordings are contaminated by the stimulus waveform and EMG signals. With this higher charge per phase, the axonal excitement was so great that it induced muscle twitching at a greater degree than in Figure 4, so the EMGs highly contaminated the signal.

### 3.2. CNAPs and Recruitment Plots: Intact vs. Transected Nerves

To assess the influence of EMG contamination on the calculation of recruitment plots, we compare the recruitment for Rat3 pre and post nerve transection. The left nerve branches (T, P, and S) were distally severed (~5 mm distally from the recording cuff electrodes). Figure 6 shows the CNAPs recorded from the three branches while stimulating the left sciatic nerve of the animal while the nerves did and did not innervate the muscles. The CNAPs of the tibial branch follow the regular trend of cresting and troughing all within 1 ms after the stimulation onset. There is a slight difference in amplitude between the curves with and without nerve transection; the influence of the EMGs reduces the amplitudes, but the relative trend with regard to an increasing pulse width is the same. The CNAP shape in the peroneal and sural branches has been much more affected by the influence of the EMGs since their CNAP signal was weaker. For the recruitment plots from these data, in the tibial nerve, the AUC was calculated from the onset of the CNAPs, i.e., the first positive phase to the end of the negative phase, while just the positive phase of the recruitment curve was used for the peroneal and sural branches in the intact nerves since the EMGs corrupted the negative phase. Figure 7 shows the recruitment curves plotted in one graph. The solid lines represent the intact nerve, and the dashed lines represent the transected nerve. Both recruitment curve sets were normalized to the respective maximum value for the transected and non-transected cases. It is clear that the overall recruitment shapes are very similar between the two cases. The onset of recordable activation due to stimulation occurs near 30 μA for the tibial nerve and 40 μA for the other branches. Evaluating the recruitment curves can also give a rough estimate of where the intrafascicular electrode might be placed and thus relate the selectivity of the neural implant. In this case, the stimulating electrode seems to be further inside the tibial nerve branch than the others.

### 3.3. Activation of the Sciatic Nerve Branches and Placement of the Electrode 

One of the objectives of this study is to determine the position of the intrafascicular electrodes via evaluation of their recruitment curves calculated using the measured CNAPs.

Examples of the CNAPs and recruitment plots from Rat1 are given in Figure 8 and Figure 9.

From the plot, we can concur that at the left sciatic nerve, the stimulation site is positioned nearest to the tibial fascicle since relatively no activation is present in the other nerve branches. However, due to the relatively small resultant CNAP voltage value (i.e., an order of magnitude smaller than the others), there is a lot of noise in the CNAP signals and in the recruitment curve as well. Since the onset of stimulation (around 30 μA) is similar to that in the other data (Rat3), the source of the low-voltage recording would seem to stem from the cuff recording rather than the interfascicular vs. intrafascicular placement of the hd-TIME. Figure 9 shows the stimulation response of the right sciatic nerve of Rat1. The CNAP activation in the right side of the nerve is more robust and similar in amplitude to Rat2 and Rat3. In this case, the peroneal activation is prominent. This concludes that the electrode is more likely situated in the peroneal region (Figure 10). Also, part of the CNAPs has been engulfed by the stimulation artifact. This happens when the stimulation pulse width is too broad or there is a smaller separation between the stimulation and recording sites. Figure 10 shows estimates of the relative placement of the stimulating electrode array in the sciatic nerve of the rats. The figures are drawn to scale.

### 3.4. Surgery and Implantation

Surgery and implantation in rats may not be an ideal model for validating peripheral neural electrodes meant for humans. A suitable animal model would be one which has peripheral anatomy similar to that of a human [53]. It would also give a closer prediction of the functional bio-reaction to implantation post implantation. But for the early stages of the development and fabrication of a novel neural interface, small animal models, i.e., rats, are a better choice to obtain quick results due to the ease of surgery. This is important for the early design and implant assessment of the interface. Therefore, we investigated the measurement of the efficacy of the neural interface stimulation performance in small animal models, keeping in mind the procedures needed for long-term implantation. Some of the CNAP recordings were contaminated by the EMG signals. This could have been avoided by severing the nerve from the innervated muscle distal to the cuff electrodes. However, this method would be useless during chronic studies. So, knowing when and where contamination happens and how to distinguish the CNAPs from EMGs during offline processing are also the objectives of this study. An optimal distance for the placement of the stimulating to recording electrodes in a rat model is around 16 mm with a stimulus waveform of a pulse width between 50 µs and 100 μs, according to our results.

## 4. Conclusions

The future of bidirectional neural interfaces depends upon their ability to decode the motor intention during stimulation and recording and relay it to the brain through the afferent fibers in the quickest possible way. This study showed that the current neural interface can stimulate selectively, albeit in the passive stage with limited capabilities. Further research is needed to evaluate a fully functional hd-TIME and exploit all its functions, along with custom-built electronics. 

## Figures and Tables

**Figure 1 micromachines-15-00280-f001:**
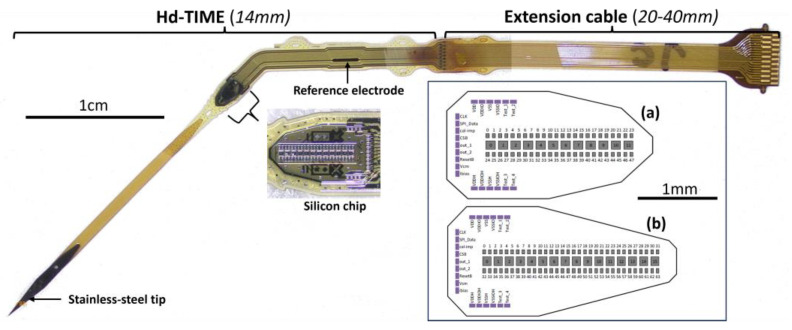
The hd-TIME probe comprises a polyimide ribbon with a stainless-steel micro-knife attached at the tip for easy implantation. The main probe body, which will reside inside the nerve, consists of a double-sided back-to-back structure, with each side containing a CMOS-compatible silicon chip with stimulation and recording sites encapsulated within a biocompatible and flexible support matrix. The hd-TIME has two variants. The short chip (**a**) has 12 stimulation and 48 recording sites. The long version (**b**) has 16 stimulation and 64 recording sites.

**Figure 2 micromachines-15-00280-f002:**
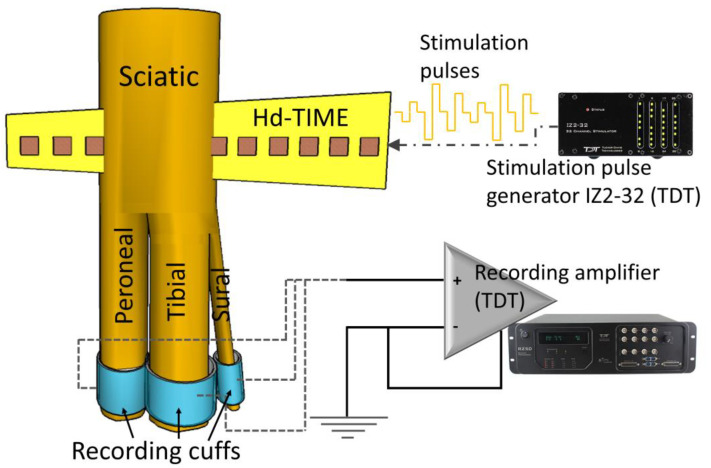
Biphasic cathode-leading rectangular pulse trains, ranging from 2 µAmp to 650 µAmp in pseudorandom order, pulsewidth of 50 µs and 100 µs, stimulation frequency of 5 Hz, were delivered to the hd-TIME using the IZ2-32 stimulator. The CNAPs were recorded using the cuff electrodes placed on each of the sciatic nerve branches, viz. T, P, and S. These signals were pre-amplified and recorded for post-processing.

**Figure 3 micromachines-15-00280-f003:**
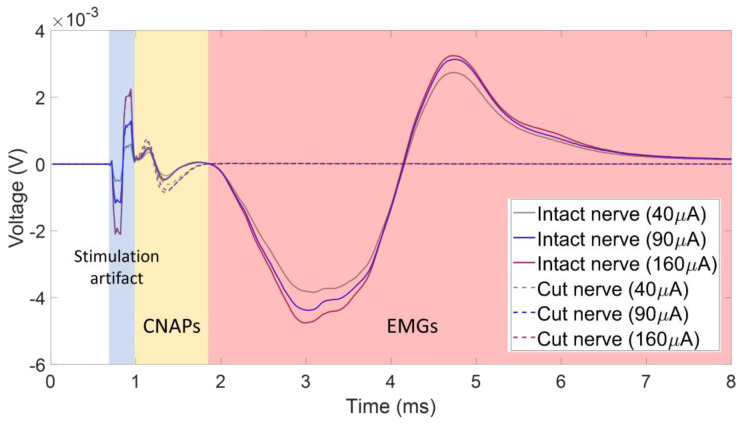
Example of CNAPS obtained from the tibial nerve branch without and with nerve transection at the distal site, when the sciatic nerve was stimulated using an hd-TIME in Rat3. The pulse train had a 100 μs pulse width and various pulse amplitudes, shown in the graph. The disappearance of the EMG after transection should be noted. The simulation-to-recording distance for this rat was about 16 mm.

**Figure 4 micromachines-15-00280-f004:**
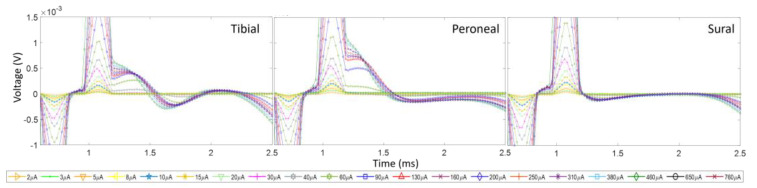
CNAPs obtained when the right sciatic nerve of Rat1 was stimulated with pw = 100 µs with various stimulation amplitudes. These results show the effect of the stimulus artifact obscuring the CNAPS due to an inadequate stimulation-to-recording distance. There was a 13 mm distance between the stimulating and recording electrodes. The recording was taken using cuff electrodes on distal branches of the tibial, peroneal, and sural nerves.

**Figure 5 micromachines-15-00280-f005:**
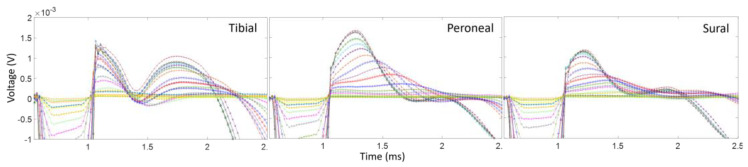
CNAPs obtained when the left sciatic nerve of Rat2 was stimulated with pw = 500 µs with various stimulation amplitudes. These results show the effect of a longer pulse width on nerve recording even while maintaining close to 16 mm distance between the stimulating and the recording electrodes. The recording was taken using cuff electrodes on distal branches of the tibial, peroneal, and sural nerves.

**Figure 6 micromachines-15-00280-f006:**
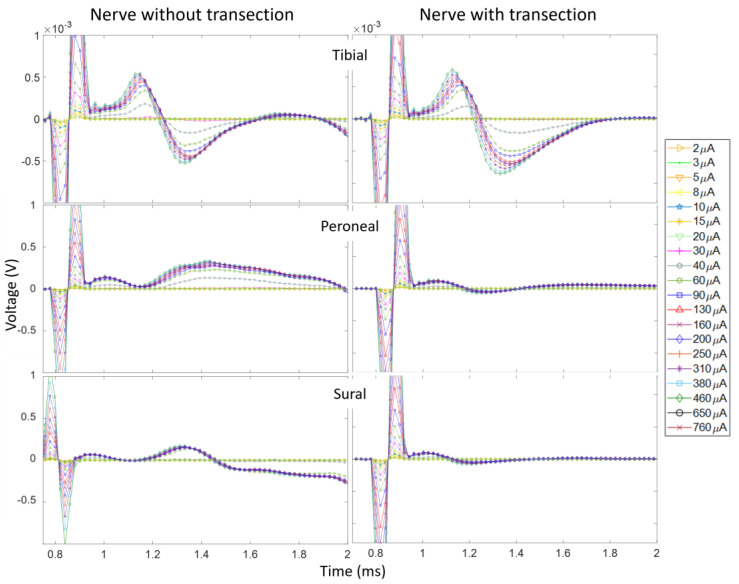
CNAPs recorded from Rat3 before and after nerve transections. The stimulation amplitudes ranged from 2 to 760 μA and the pulse width was 50 μs.

**Figure 7 micromachines-15-00280-f007:**
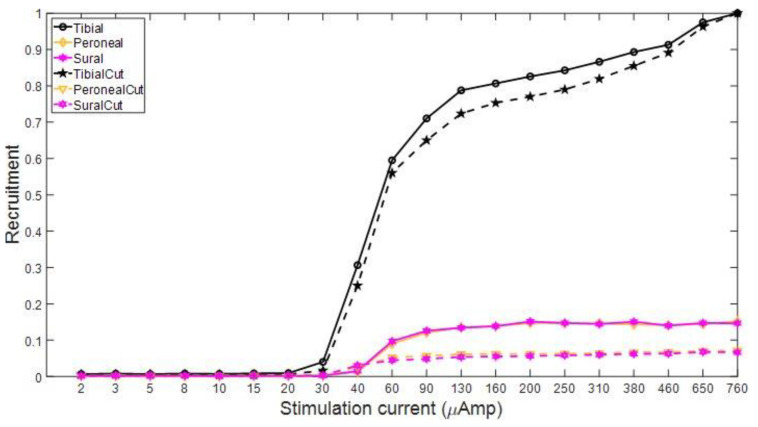
Recruitment curves corresponding to CNAP curves in Figure 6. The dashed lines are after nerve transection and the solid lines are before. Recruitment curves were normalized to the maximum value in the cut and uncut tibial nerve CNAPs, respectively.

**Figure 8 micromachines-15-00280-f008:**
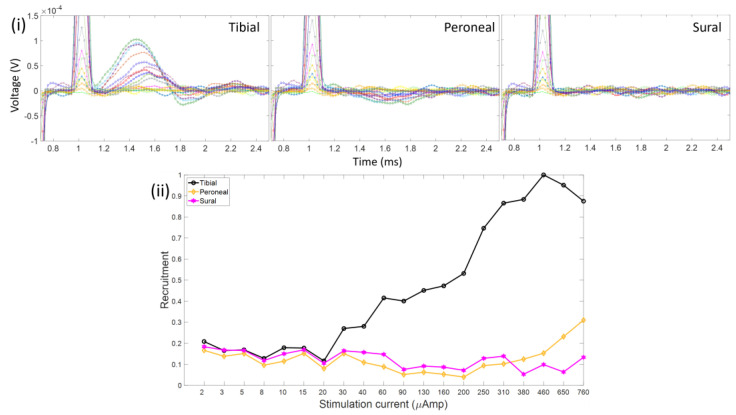
(**i**) CNAPs measured from stimulation (pw = 100 µs) in the left sciatic nerve for Rat1. Notice the relatively small voltage amplitude and selective activation of the tibial nerve branch. (**ii**) Resultant recruitment plots of the three respective nerve branches. A lot of noise is present in this measurement due to a low signal-to-noise ratio.

**Figure 9 micromachines-15-00280-f009:**
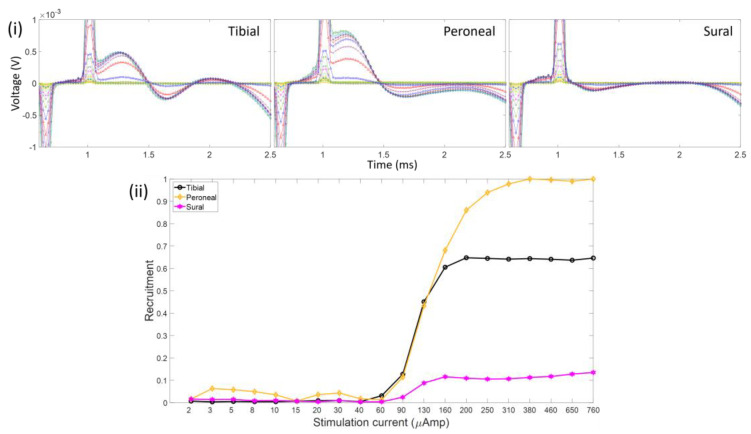
(**i**) CNAPs measured from stimulation (pw = 50 µs) in the right sciatic nerve of Rat1. Notice the greater response of recruitment in the peroneal nerve. (**ii**) Resultant recruitment plots of the three respective nerve branches.

**Figure 10 micromachines-15-00280-f010:**
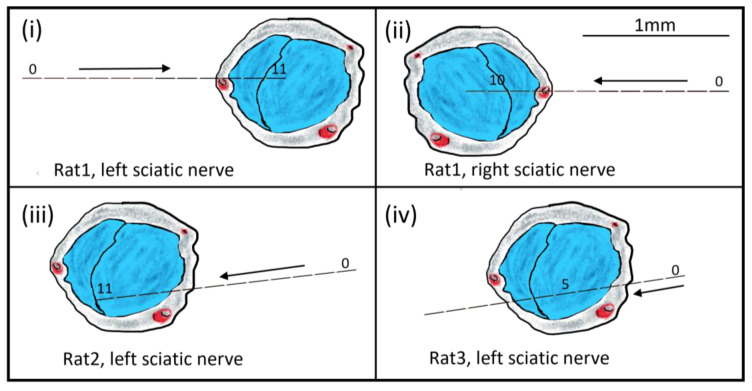
Schematics (**i**–**iv**) depicting hd-TIME estimation of implantation direction and placement of the stimulation electrodes inside the proximal region of the sciatic nerve. The cross-section of the sciatic nerve shows the peroneal branching (the smaller section) and the tibial region. The sural fascicle starts branching slightly distal from the implant region and has not been shown here. The numbers on the electrode inside the nerve distinguish the active electrode. The electrode array and nerves were drawn to scale.

## Data Availability

Data is available upon request due to ethical restrictions.

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
