# Peer review of "Using Compound Neural Action Potentials for Functional Validation of a High-Density Intraneural Interface: A Preliminary Study"

_micromachines, 2024, doi:10.3390/mi15020280_

Round 1
Reviewer 1 Report
Comments and Suggestions for Authors
The manuscript describes acute in vivo testing of the new type of intranueral electrode, hd-TIME, using Compound Neural Action Potentials (CNAPs) as a means to assess stimulation capabilities. Authors describe their methodology and results in sufficient detail, and the experiment is well executed. However, the presentation of the results, abstract, and conclusions sections are not justified. I believe this article could be published after revisions in the text that address the presentation of the data and reformulation of the main scientific goals.
The stated hypotheses that the authors are aiming to test is that "the nerve has a certain amount of flexibility to withstand small injuries and still function normally" (line 19) and verify the methodology of measuring CNAPs for stimulation using a high-density electrode that was designed with a large animal model in mind (humans included). I would argue that the first part can't be verified in the acute setting, and a chronic study is necessary. This is especially important since the hd-TIME is a fairly wide electrode, as is demonstrated in Table 1. The second part is somewhat undermined by the fact that the authors use a passive version of the hd-TIME with reduced functionality that allows for only one stimulation site to be present in the sciatic nerve during the experiment (a single electrode is hardly a high-density interface). It is also not clear that the testing of the CNAPs methodology in rats will be transferable to other animal models as the artifact profile from EMGs will likely be different and the optimal position for recording cuffs placement will change. I would propose rewording the abstract and conclusions section around the scope of the work statement on lines 309-311. I believe it succinctly captures the major goals of this manuscript without overgeneralization.
My specific comments are below:
Line 159: Figure 1 lacks scale bars for insets a), b), and image of the Silicon chip. The location of the ground pad should be highlighted
Line 180: It is not clear what stimulation cuff authors refer to. It seems this was not used in this work
Line 193: Figure 3 is incomprehensible. There is too much data, and the legend is not sorted in either ascending or descending manner. I guess it is arranged in the pseudorandom order as in the experiment, but this makes it really hard to read. The image resolution is too low and/or the data marks are too small to read the figure. I would propose condensing this figure to illustrate the main features of the data authors would like to highlight instead of dumping all data as an "example". The voltage scale for Sural data is missing the units. All subsequent figures are missing the legend for the voltage CNAP traces.
Line 227: There is no Figure labeled 3B. I assume it refers to plots in Figure 4 that look the same as Figure 3.
Line 361: The first three sentences in the conclusions are general statements that have nothing to do with the results of this work. Please remove them. I would propose adding any conclusions regarding the use of CNAPs for evaluation of the stimulation efficiency and describing the necessity to fine-tune the recording cuff placement to avoid stimulation and EMG artifacts.
Comments on the Quality of English LanguageI would propose to reword several sentences to improve clarity:
lines 150,154,171, 297
Author Response
The authors thank the reviewers for taking the time to highlight the problems. The issues were noted and have been addressed to improve the quality of the manuscript. We have added texts and figures in the necessary places to address the concerns of the reviewers.
My specific comments are below:
Line 159: Figure 1 lacks scale bars for insets a), b), and image of the Silicon chip. The location of the ground pad should be highlighted.
Addressed the issues.
Line 180: It is not clear what stimulation cuff authors refer to. It seems this was not used in this work
Addressed the issue.
Line 193: Figure 3 is incomprehensible. There is too much data, and the legend is not sorted in either ascending or descending manner. I guess it is arranged in the pseudorandom order as in the experiment, but this makes it really hard to read. The image resolution is too low and/or the data marks are too small to read the figure. I would propose condensing this figure to illustrate the main features of the data authors would like to highlight instead of dumping all data as an "example". The voltage scale for Sural data is missing the units. All subsequent figures are missing the legend for the voltage CNAP traces.
The voltage scale for every plot is scaled to the same value for the corresponding Tibial nerve (the left most plot). All the plots have been redrawn with higher resolution.
Line 227: There is no Figure labeled 3B. I assume it refers to plots in Figure 4 that look the same as Figure 3.
Problem has been addressed.
Line 361: The first three sentences in the conclusions are general statements that have nothing to do with the results of this work. Please remove them. I would propose adding any conclusions regarding the use of CNAPs for evaluation of the stimulation efficiency and describing the necessity to fine-tune the recording cuff placement to avoid stimulation and EMG artifacts.
The results section has been revised thoroughly.
Comments on the Quality of English Language
I would propose to reword several sentences to improve clarity:
lines 150,154,171, 297
We have tried our best to improve the level of English.
Reviewer 2 Report
Comments and Suggestions for Authors
The authors describe a method to quantify the stimulation efficacy of neural implants by recording so-called compound nerve action potentials (CNAPs).
While the method and its application is appreciated, I would suggest to the authors to greatly improve the presentation of the data and better structure the abstracts and introduction to make sure to appeal to a broader audience than the current state of the manuscript permits.
1. Arranging CNAP response color scheme in pseudo random order of stimulus uAmp size does not help interpret the plots, it hinders it. I would use a perceptually linear colormap not containing white colors, and plot the curves accordingly. This would allow to naturally follow the curves' progressions in terms of color magnitude (which follows uAmp magnitude).
2. Most of the useful information on the technique is actually contained in the recruitment plots. I think Figures could be greatly collapsed. Most CNAP curves seem to be reasonably either phase-shifted or upscaled versions of each other for a given pulse width and probed region. One such detailed plots with the 10+ curves should suffice to convey what is investigated, and all the following plots could show the average waveform per stimulus width/per region, and then provide all 3 recruitment plots in a single figure. These recruitment figures would then help interpret (with the help of arrows) how figure 8 is put together.
While this is a matter of taste, I think it would help with the clarity of the article, which I found hard to follow. In general, it is not sufficiently clear at first sight from the Abstract and the Introduction what problem is tackled, why, and what the authors propose to do. It is hard to see why subsections 1.1, 1.2 and 1.3 have logical consistency and are necessary to the appreciation of the article.
Comments on the Quality of English LanguageThe language is hard to follow with many repetitions, many typos (e.g "Also, nerves hold a certain degree of stretchiness due to present 87 the Fontana bands") and many awkward sentences. I would suggest a native speaker or an experienced writer to review the language.
In general, paragraphs seem to randomly follow each other without taking care of introducing the concept, fleshing it out, and finishing it on a topic that justifies the start of the next paragraph.
Author Response
The authors thank the reviewers for taking the time to highlight the problems. The issues were noted and have been addressed to improve the quality of the manuscript. We have added texts and figures in the necessary places to address the concerns of the reviewers.
The authors describe a method to quantify the stimulation efficacy of neural implants by recording so-called compound nerve action potentials (CNAPs).
While the method and its application is appreciated, I would suggest to the authors to greatly improve the presentation of the data and better structure the abstracts and introduction to make sure to appeal to a broader audience than the current state of the manuscript permits.
- Arranging CNAP response color scheme in pseudo random order of stimulus uAmp size does not help interpret the plots, it hinders it. I would use a perceptually linear colormap not containing white colors, and plot the curves accordingly. This would allow to naturally follow the curves' progressions in terms of color magnitude (which follows uAmp magnitude).
The issue has been addressed.
- Most of the useful information on the technique is actually contained in the recruitment plots. I think Figures could be greatly collapsed. Most CNAP curves seem to be reasonably either phase-shifted or upscaled versions of each other for a given pulse width and probed region. One such detailed plots with the 10+ curves should suffice to convey what is investigated, and all the following plots could show the average waveform per stimulus width/per region, and then provide all 3 recruitment plots in a single figure. These recruitment figures would then help interpret (with the help of arrows) how figure 8 is put together.
We have added and simplified the CNAP plots and have recruitment plots for the relevant ones. We agree too many plots of similar types can be confusing.
While this is a matter of taste, I think it would help with the clarity of the article, which I found hard to follow. In general, it is not sufficiently clear at first sight from the Abstract and the Introduction what problem is tackled, why, and what the authors propose to do. It is hard to see why subsections 1.1, 1.2 and 1.3 have logical consistency and are necessary to the appreciation of the article.
We truly agree with the suggestion. Although our goal was to show the changes in CNAPs characteristics for different types of implants.
Comments on the Quality of English Language
The language is hard to follow with many repetitions, many typos (e.g "Also, nerves hold a certain degree of stretchiness due to present 87 the Fontana bands") and many awkward sentences. I would suggest a native speaker or an experienced writer to review the language.
In general, paragraphs seem to randomly follow each other without taking care of introducing the concept, fleshing it out, and finishing it on a topic that justifies the start of the next paragraph.
We have tried our best to improve the level of English.